# The use of non-functional clonotypes as a natural calibrator for quantitative bias correction in adaptive immune receptor repertoire profiling

Anastasia O Smirnova[1,2], Anna M Miroshnichenkova[2,3], Yulia V Olshanskaya[2,3], Michael A Maschan[3], Yuri B Lebedev[2,4], Dmitriy M Chudakov[1,2,4,5], Ilgar Z Mamedov[2,4], Alexander Komkov[2,3]*

[1]Skolkovo Institute of Science and Technology, Moscow, Russian Federation; [2]Department of Genomics of Adaptive Immunity, Shemyakin-Ovchinnikov Institute of Bioorganic Chemistry, Moscow, Russian Federation; [3]Dmitry Rogachev National Medical Research Center of Pediatric Hematology, Oncology and Immunology, Moscow, Russian Federation; [4]Pirogov Russian National Research Medical University, Moscow, Russian Federation; [5]Abu Dhabi Stem Cells Center, Abu Dhabi, United Arab Emirates

**Abstract** High-throughput sequencing of adaptive immune receptor repertoires is a valuable tool for receiving insights in adaptive immunity studies. Several powerful TCR/BCR repertoire reconstruction and analysis methods have been developed in the past decade. However, detecting and correcting the discrepancy between real and experimentally observed lymphocyte clone frequencies are still challenging. Here, we discovered a hallmark anomaly in the ratio between read count and clone count-based frequencies of non-functional clonotypes in multiplex PCR-based immune repertoires. Calculating this anomaly, we formulated a quantitative measure of V- and J-genes frequency bias driven by multiplex PCR during library preparation called Over Amplification Rate (OAR). Based on the OAR concept, we developed an original software for multiplex PCR-specific bias evaluation and correction named iROAR: immune Repertoire Over Amplification Removal (https://github.com/smiranast/iROAR). The iROAR algorithm was successfully tested on previously published TCR repertoires obtained using both 5' RACE (Rapid Amplification of cDNA Ends)-based and multiplex PCR-based approaches and compared with a biological spike-in-based method for PCR bias evaluation. The developed approach can increase the accuracy and consistency of repertoires reconstructed by different methods making them more applicable for comparative analysis.

*For correspondence: alexandrkomkov@yandex.ru

Competing interest: The authors declare that no competing interests exist.

## Editor's evaluation

This paper describes a newly developed, publicly available algorithm (iROAR) that was tested on pre-exisiting datasets and is of interest to T and B cell immunologists who perform repertoire analysis via multiplex PCR based techniques. iROAR utilises naturally occurring non-functional sequences to improve and partially correct the amplification bias inherent in multiplex PCR based sequencing technologies.

## Introduction

Adaptive immune receptor (TCR – T-cell receptor and BCR – B-cell receptor) repertoire is usually defined as a set of TCR or BCR sequences obtained from an individual's blood, bone marrow, or specific lymphocyte population. Reflecting the T/B cell's clonal composition, the repertoire is characterized

by a high degree of specificity for each individual and substantial variation in clone frequencies. The accuracy of both sequences and frequencies of TCR/BCR genes in the obtained repertoire is essential to receiving the correct biological information from immune repertoire analysis.

High-throughput sequencing (HTS) of adaptive immune receptor repertoires is widely used in immunological studies (reviewed in *Minervina et al., 2019*) for the investigation of immune response to vaccines (*Minervina et al., 2021*; *Pogorelyy et al., 2018*; *Sycheva et al., 2022*), tumor-infiltrating lymphocytes (*Gee et al., 2018*; *Goncharov et al., 2022*; *Oliveira et al., 2021*), new therapeutic agents (*Huang et al., 2019*; *Wang et al., 2018*; *Wilson et al., 2022*), leukemia clonality, and minimal residual disease monitoring (*Brüggemann et al., 2019*; *Komkov et al., 2020*; *Nazarov et al., 2016*; *Tirtakusuma et al., 2022*; *Wood et al., 2018*). HTS-based methods for immune repertoire profiling use either RNA or DNA as a starting material and, in most cases, use PCR for the selective enrichment of receptor sequences. DNA-based methods generally use two-sided multiplex PCR with primers annealing to multiple V- and J-genes of the rearranged receptor (*Brüggemann et al., 2019*; *Komkov et al., 2020*; *Robins et al., 2009*). RNA-based methods start with cDNA synthesis, usually with TCR/BCR C(constant)-genes specific oligonucleotides, followed by one-side multiplex amplification with a set of V-gene specific primers and a universal C-gene specific primer (*Wang et al., 2010*). Alternatively, two universal primers are used for amplification if an artificial sequence is added to the 5′ end during synthesis using a template-switch (5′-Rapid Amplification of cDNA Ends [RACE]; *Mamedov et al., 2013*) or ligation (*Oakes et al., 2017*). DNA-based methods protect the repertoire from gene transcription bias and provide more comprehensive results (*Barennes et al., 2020*) which include most non-functional (out-of-frame) as well as functional (in-frame) rearrangements but produce high-amplification bias in the course of multiplex PCR. Additionally, each T/B cell contains a single DNA copy (i.e. two target strands) of the receptor molecule in contrast to tens of single-stranded RNA copies. RNA-based methods using 5′-RACE or ligation are characterized by the lowest PCR bias as they need a single primer pair for the amplification. However, the low efficiency of adding a universal oligo to the 5′ end makes its sensitivity comparable to or even lower than DNA-based methods. The compromise between these two approaches is the RNA-based method with a one-side multiplex that has moderate amplification bias yet sufficient sensitivity (*Ma et al., 2018*). Most bias in one-side multiplex RNA-based approaches could be removed by using unique molecular identifiers (UMIs; *Ma et al., 2018*). Unfortunately, for DNA-based methods, efficient incorporation of UMIs into the initial molecule before PCR is still challenging. The only method for DNA multiplex bias correction (*Carlson et al., 2013*) is undirected and cost-ineffective due to the utilization of an expensive synthetic spike-in control repertoire. Here, we propose an orthogonal solution for this challenge: the first fully computational algorithm for amplification bias detection and correction in adaptive immune receptor repertoires named iROAR (immune Repertoire Over Amplification Removal).

## Results

### The rationale for the Over Amplification Rate measure

Since out-of-frame TCR/BCR rearrangements do not form a functional receptor, they are not subjected to any specific clonal expansions and selection (*Murugan et al., 2012*). Being a passenger genomic variation, they change their initial (recombinational) clonal frequencies just randomly following the frequency changes of the second functional (in-frame) TCR/BCR allele present in the same T/B cell clone. According to the TCR/BCR loci rearrangement mechanism, the formation of in-frame and out-of-frame allele combinations in the same cell is also a stochastic and independent process in terms of V- and J-genes frequency. It leads to the conclusion that V- and J-gene frequencies among out-of-frame rearrangements must be sufficiently stable and must be equal to the initial recombination frequencies despite repertoire changes caused by various immune challenges (*Figure 1*). Thus, reproducible deviation of out-of-frame V- and J-gene frequencies (for the same multiplex PCR primer set) from the initial recombinational frequencies observed in the sequenced repertoire dataset is a result of artificial aberration caused by PCR amplification rather than immune repertoire evolution. Thus out-of-frame clonotypes can be considered a natural calibrator that can be used to measure amplification bias and quantitatively correct immune repertoire data.

Formulating this observation, we developed the Over Amplification Rate (OAR) measure, which we define as a ratio of the observed and expected frequency of a V- (OAR[Vi]) or a J-gene (OAR[Ji])

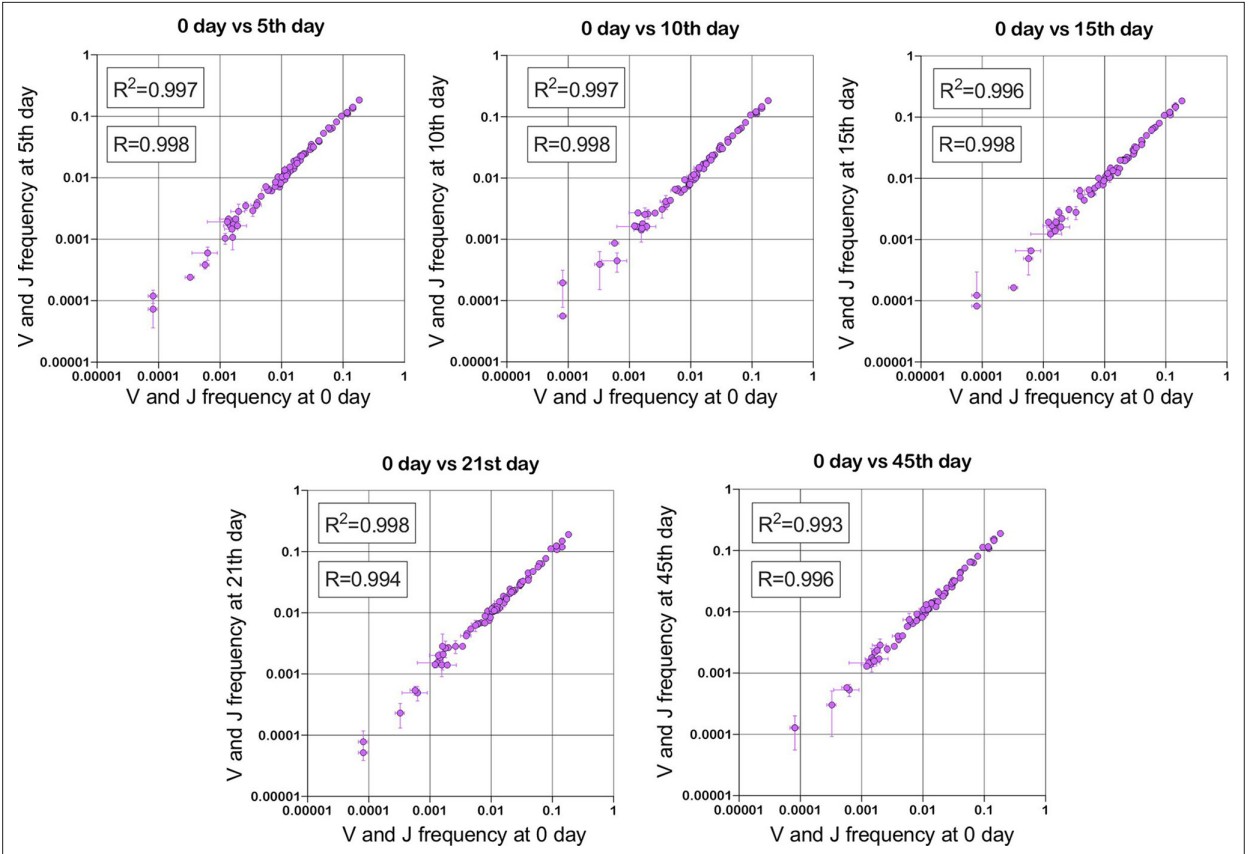

**Figure 1.** Stability of TRBV and TRBJ genes frequencies calculated based on unique out-of-frame rearrangements after Yellow fever vaccination (model of acute viral infection). Out-of-frame clonotypes for frequencies calculation were extracted from low-biased 5' Rapid Amplification of cDNA Ends (RACE) TRB repertoires of PBMC (peripheral blood mononuclear cells) samples obtained in two replicates for six time points: 0, 5, 10, 15, 21, and 45 d after YFV (Yellow Fever Vaccine) injection (donor M1, SRA accession number PRJNA577794 *Minervina et al., 2020*).

The online version of this article includes the following source data for figure 1:

**Source data 1.** XLSX table.

among identified out-of-frame rearrangements. Observed frequency represents a value calculated as read counts (RCs) for each V- and J-gene (related to out-of-frames) divided by the sum of all out-of-frame clones RC in the obtained repertoire sequencing dataset. The expected frequency is a value before amplification calculated as a number of unique out-of-frame clones (UCN) having each V- or J-gene divided by the total number of unique out-of-frame clones in the repertoire. At the final stage, each OAR is normalized by dividing by the average OAR.

$$OAR\left(Vi\right) = \frac{\frac{RC(Vi)}{\sum_1^N RC(Vi)}}{\frac{UCN(Vi)}{\sum_1^N UCN(Vi)}}$$

$$OAR\left(Ji\right) = \frac{\frac{RC(Ji)}{\sum_1^N RC(Ji)}}{\frac{UCN(Ji)}{\sum_1^N UCN(Ji)}}$$

OAR value tends to be equal to 1 under ideal conditions (low or no amplification bias). It deviates from 1 as amplification bias increases in line: 5'-RACE with a single universal primer pair, one-side multiplex PCR (VMPlex), and two-side multiplex PCR (VJMPlex; *Figure 2*).

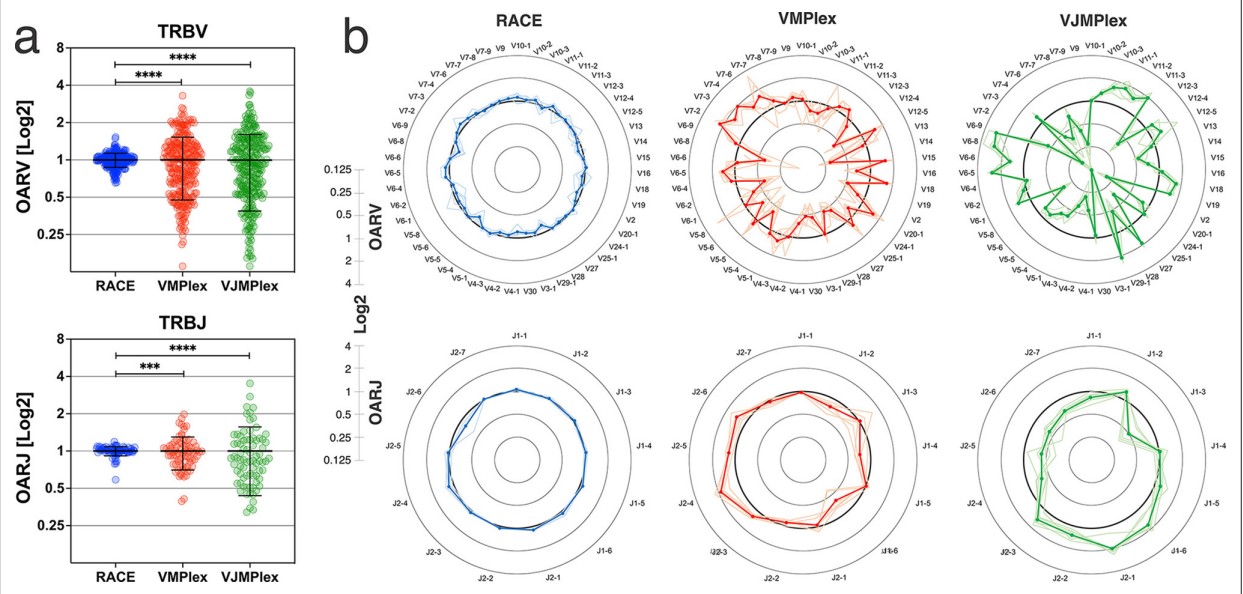

**Figure 2.** Dependence of the Over Amplification Rate on the TCR repertoire profiling method.

(**a**) Comparison of Over Amplification Rate (OAR) values variances for TRB repertoires obtained with 5'-Rapid Amplification of cDNA Ends (RACE), one-side multiplex (VMPlex), and two-side multiplex (VJMPlex) PCR. The Levene's test was performed to compare OAR variances: ****p<0.0001 and ***p<0.001. The bar and whiskers indicate a mean and SD. (**b**) Average (bold lines) OAR values for TRBV and TRBJ genes in repertoires obtained with 5'-RACE, one-side multiplex (VMPlex), and two-side multiplex (VJMPlex) PCR. Pale lines illustrate OARs of individual repertoires. Datasets: six repertoires for RACE from PRJNA847436 (*Sycheva et al., 2022*), six repertoires for VMPlex from PRJNA427746 (*Ma et al., 2018*), and six repertoires for VJMPlex from 27483#.XpCuQ1MzZQI (zenodo.org; *Weinberger et al., 2015*).

The online version of this article includes the following source data and figure supplement(s) for figure 2:

**Source data 1.** XLSX table.

**Figure supplement 1.** Comparison of population V/J segment frequencies with individual frequencies obtained using different TCR repertoire profiling methods.

## The versatility of OAR measure

OAR measurement is a universal approach and can be applied to different types of immune repertoire data. To demonstrate this versatility, we calculated OAR values for low-biased (5' RACE) repertoires of different adaptive immune receptor chains obtained from bulk human PBMC: TCR alpha (TRA), TCR beta (TRB), and BCR heavy chains (*Figure 3a*). The results show that OARs for both TCR and BCR repertoires obtained by 5' RACE are close to 1 and stay within the range of 0.5–2, which is much narrower than OAR for multiplex PCR-based repertoires (see main text *Figure 2*).

We also analyzed OARs for low-biased (5' RACE) TCR repertoires of different T cell subpopulations, including T-helper, cytotoxic, central memory, effector memory, and naïve T cells. As shown in *Figure 3*, the OAR values demonstrate much less differences between analyzed T cell types then between RACE and multiplex PCR and are close enough to 1 similarly to the repertoire of bulk T cell mix obtained from PBMC.

Herewith, the variance of IGHV's OARs compared TCRs' and the variance of TCR subpopulations OARs compared PBMCs' is slightly higher. This phenomenon may be linked to well-known differences in clonal expansion intensities of B/T-cell subsets which can affect indirectly the OAR values. However, the proof of this hypothesis demands separate deep analysis which is beyond the main focus of this research.

Despite it, our results demonstrate that OAR is a sufficiently universal measure of repertoires and can be applied to most adaptive immune receptors and cell types.

## Factors affecting OAR measure accuracy

In the case of insufficient sequencing coverage, high-PCR bias can lead to the dramatic loss of clones and thus an incorrect measurement of V- and J-genes frequencies. In this instance, for the majority of

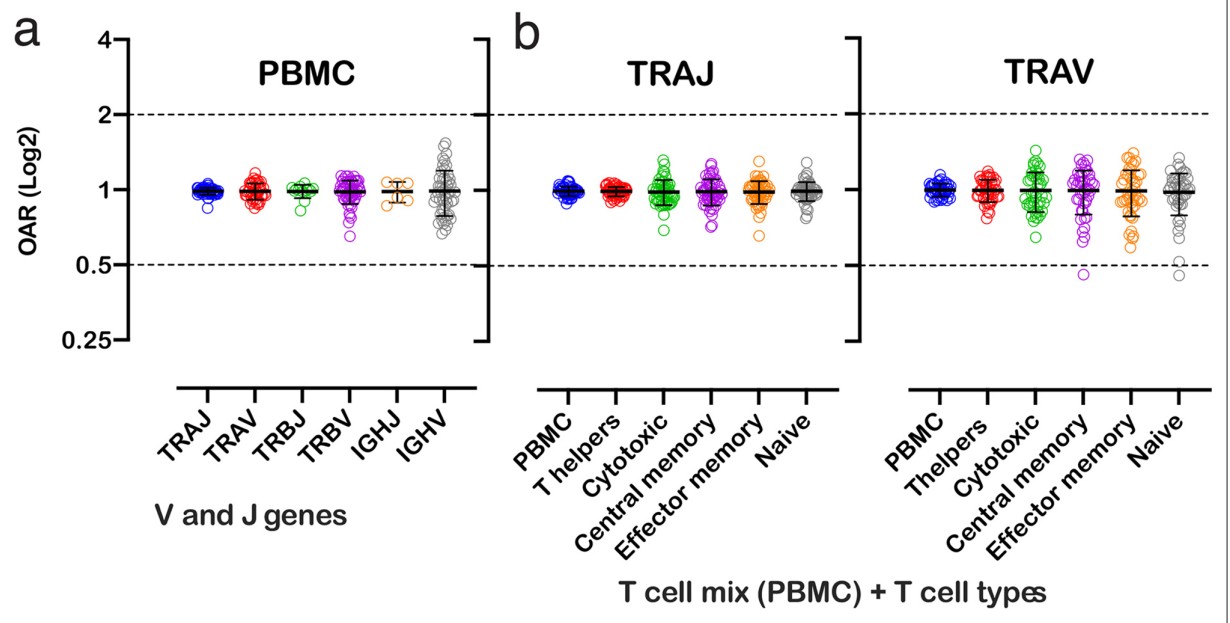

**Figure 3.** Over Amplification Rate of low-biased TCR and BCR repertoires in different lymphocyte subsets.
(**a**) Distribution of Over Amplification Rates of V- and J-genes in Rapid Amplification of cDNA Ends (RACE)-based repertoires of TCR and BCR (the empty dots represent average OARs among TCR repertoires: SRA accession numbers: PRJNA577794, PRJNA316572, PRJEB27352, and BCR repertoires: SRA accession number: PRJNA297771 and PRJNA494572). (**b**) Over Amplification Rates of V- and J-genes of TCR alpha chains in RACE-based repertoires of different types of T-cells (donors M1 and P30, 45 d after booster vaccination, SRA accession number PRJNA577794; *Minervina et al., 2020*). The bar and whiskers indicate a mean and SD.

The online version of this article includes the following source data for figure 3:

**Source data 1.** XLSX table.

V- and J-genes, the population frequencies can approximate the real frequencies better than multiplex repertoire-based ones (*Figure 2—figure supplement 1*). If upon comparison samples' UCN-based frequencies significantly differ from the average frequencies calculated for the population (i.e. exceeds 99% CI), OAR calculation should be based on the latter.

Also, the balance of V- and J-genes frequencies can be disrupted by accidentally arisen abnormally large non-functional clonotypes generated in the course of abnormal clonal expansion in various lymphoproliferative disorders or stochastic spike in normal lymphocyte population. To reduce the impact of this anomaly on OAR value, the top clone of each V- and J-gene containing subgroups must be excluded from OAR calculation. Since V- and J-specific bias affects all clones non-selectively, the remaining large part of clones after top clones exclusion should be still representative for PCR bias calculation. As shown in *Figure 4a*, the exclusion of one top clonotype from OAR calculation for RACE-based TRB repertoire is enough to restore OAR calculation accuracy for TRBV2, TRBV5-6, TRBV7-9, and TRBV11-3. The further top clones exclusion has no significant effect on OAR values.

Another aspect impacting the accuracy of OAR calculation is the low-sequencing coverage of the TCR/BCR repertoire. The ratio of total RCs and the sum of unique clone counts can affect OAR value despite PCR bias solely because of the mathematical properties of the OAR formula. In the extreme case, the OAR value (OAR = 1) for V- and J-genes represented in a single out-of-frame clone with only one read will not reflect the real amplification bias. To address this issue, we analyzed the OAR calculation error as a function of the number of reads per clone used for the OARs evaluation (*Figure 4b*). For this purpose, we performed a serial down-sampling of TCR datasets generated by RACE and two-side multiplex PCR and calculated OAR measurement error for each dataset portion. OAR calculated for the entire dataset was taken as a benchmark. The result shows that 1.8 (for MPlex) and 2.5 (for RACE) reads per out-of-frame clonotype are a minimal sufficient sequencing coverage to get adequate OAR values with an acceptable error rate of ~10%.

It is also important to note that errors in nucleotide sequences occurring during library preparation and sequencing could lead to an artificial increase in both in-frame and out-of-frame clone

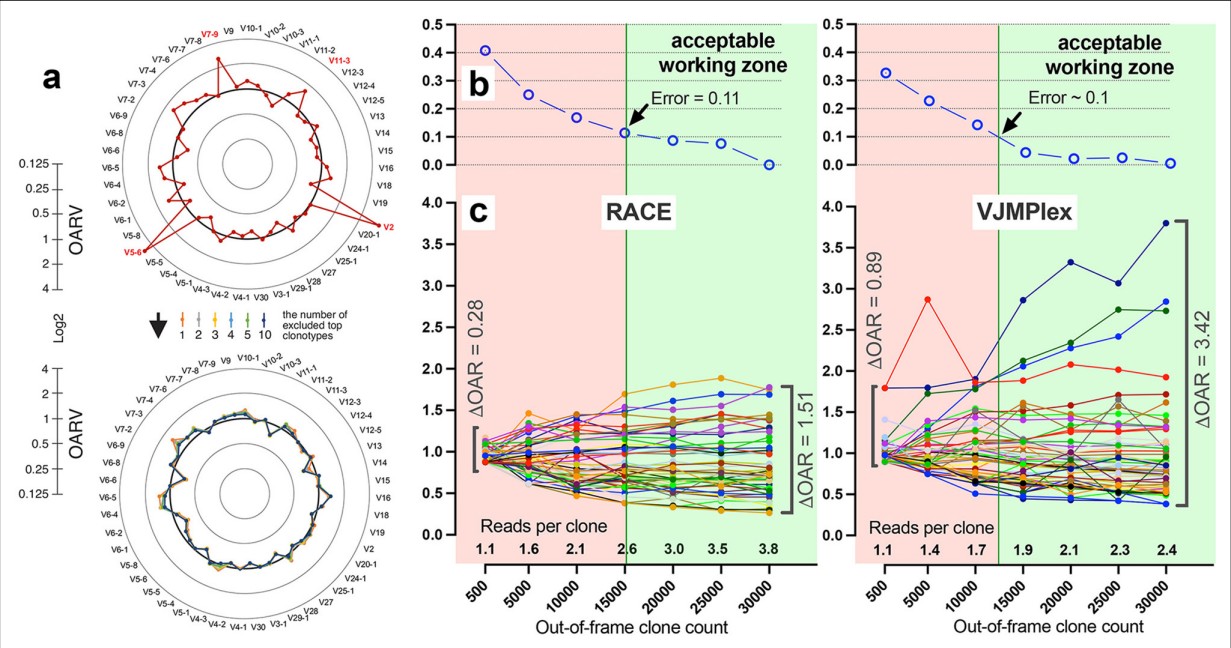

**Figure 4.** Factors impacting Over Amplification Rate (OAR) calculation accuracy. (**a**) Impact of highly proliferated top non-functional clonotype on OAR calculation accuracy in low-biased Rapid Amplification of cDNA Ends (RACE)-based TRB repertoire (Data: SRR19594184). (**b**) Impact of sequencing depth on OARs calculation error. (**c**) PCR bias independent changes of TRB V-genes OARs as a function of sequencing depth. Data: two-sided multiplex-based TRB repertoire (Data: RACE - SRR3129976, VJMPlex – SRR3129972).

The online version of this article includes the following source data for figure 4:

**Source data 1.** XLSX table.

diversity. Single nucleotide substitutions generate artificial clones as a branch of real most abundant clones inside of each in-frame and out-of-frame group independently. Single nucleotide indels lead to cross-generation artificial clones between groups: real in-frame clones generate false out-of-frame clones and vice versa. Artificial clones compromise the accuracy of both repertoire itself and OAR value. To eliminate such clones generated by single-nucleotide substitutions, we filtered them out by the VDJTOOLS software (see Methods section). To eliminate artificial clones produced by indels, we searched for in-frame and out-of-frame clone pairs which differ by one indel (Levenshtein distance = 1). If their ratio is less than 1:500, the smaller clone in pair is discarded, and its count is added to the count of the larger clone (this procedure guarantees to discard most sequencing errors present in 1 per 1000 nucleotides average).

## OAR-index

To estimate the value of immune repertoire structure disruption by amplification bias, we proposed the OAR-index, which represents the mean square deviation of OARs for each V- and J-gene from the value characteristic for repertoire with no bias (OAR = 1). OAR-index is directly proportional to the amplification bias and thus can be used for rapid estimation and comparison of immune repertoire bias. The less OAR-index is, the less PCR bias is with an ideally unbiased repertoire having OAR-index=0.

$$OAR - index = \sqrt[2]{\frac{\sum_{0}^{n}\left(OARi-1\right)^2}{n}}$$

## Using OAR for the removal of amplification bias

Normalization coefficients for each VJ combination are estimated by multiplication of corresponding V- and J-gene OARs for two-side multiplex and V-gene OAR for one-side multiplex (*Figure 2a*). The corrected RC for each clonotype with the particular V-J gene combination is obtained simply by

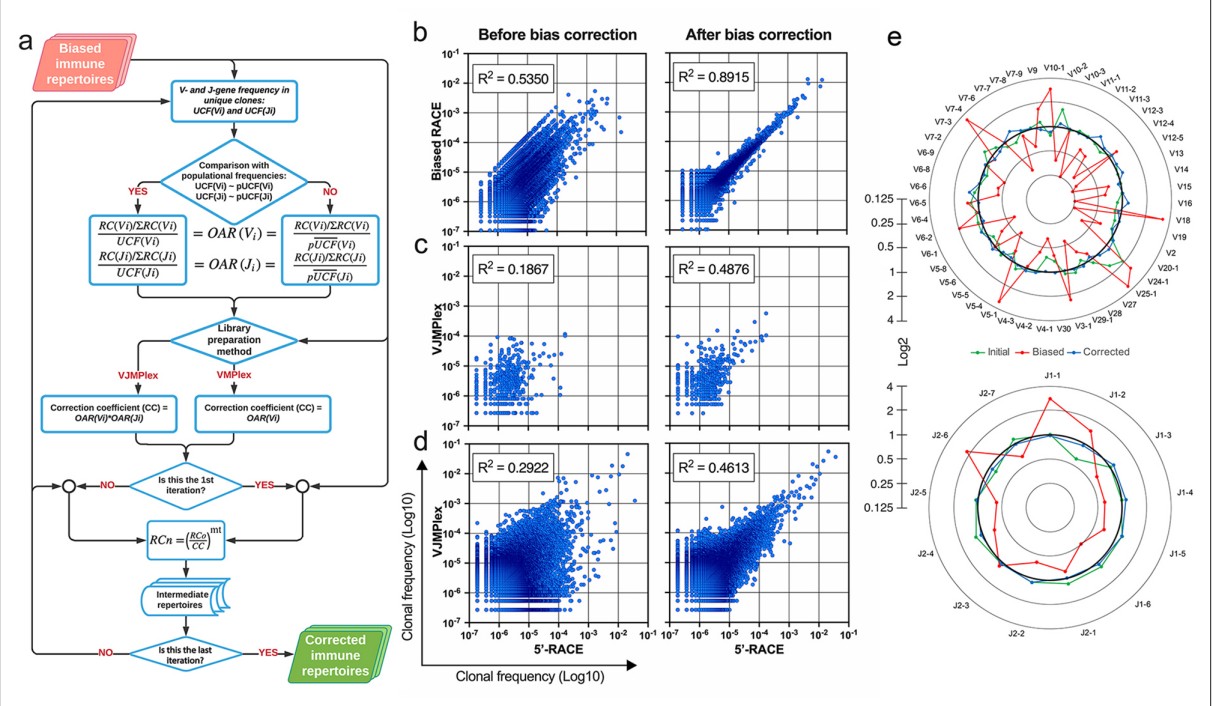

**Figure 5.** Implementation of OAR-based correction of quantitative bias in TCR repertoire.
(**a**) Flowchart of iROAR (immune Repertoire Over Amplification Removal) algorithm. UCF – Frequency calculated using unique clones counts (denominator of Over Amplification Rate [OAR]), pUCF – population UCF, RC – read count, RCn – normalized RC, RCo – observed RC, and mt – the number in the range from 0 to 1 for the iterative procedure. (**b**) Clone frequencies in the low biased 5'-Rapid Amplification of cDNA Ends (RACE)-based repertoire (ENA database, accession number ERR2869430) vs. the same repertoire with introduced artificial bias: before and after iROAR processing. (**c**) Out-of-frame (non-functional) clone frequencies in low biased 5'-RACE-based repertoire vs. two-side multiplex (VJMPlex)-based repertoire obtained for the same RNA sample (SRA database, accession numbers SRR3129976 and SRR3129972): before and after iROAR processing. (**d**) In-frame (functional) clone frequencies in low biased 5'-RACE-based repertoire vs. two-side multiplex (VJMPlex)-based repertoire obtained for the same RNA sample: before and after iROAR processing. SRA database, accession numbers SRR3129976 and SRR3129970. $R^2$ is the squared Pearson correlation coefficient. iROAR was applied only for biased repertoires: artificially biased RACE and VJMPlex. (**e**) OARV and OARJ of test 5'-RACE-based TRB repertoire (**b**) before artificial bias introduction (green dots and line), biased one (red dots and line) and corrected one by iROAR (blue dots and line).

The online version of this article includes the following source data for figure 5:

**Source data 1.** XLSX table.

dividing the observed RC by the corresponding normalization coefficient. OAR of V- and J-genes could be co-dependent, which can be a reason for overcorrection. To avoid this issue, the procedure can be recursively repeated with a modified normalization coefficient defined as described coefficient raised to the power of a number in the range from 0 to 1 (parameter 'mt'). The corrected RCs are used to estimate the real percentage of each clonotype in the repertoire. However, the all multiplex-based repertoires analyzed in actual study required just one iteration with mt = 1. A detailed flowchart of the OAR-based amplification bias correction algorithm named iROAR is shown in *Figure 5a*.

## OAR-based approach validation

The validation of OAR-based amplification bias correction was performed on the TRB dataset with in silico introduced bias generated from real (experimental) low-biased (5'-RACE) repertoire (*Figure 5b*). After correction, the OAR-index indicating general repertoire bias expectedly decreased from 1.81 to 0.76. Interestingly, the OAR independent measure $R^2$ value of in silico biased and original repertoire correlation raised from 0.5350 to 0.8915, confirming the substantial reduction of in silico introduced quantitative bias. Afterward, we tested our approach on real paired experimental datasets obtained from the same RNA sample by two different method types: 5'-RACE and multiplex PCR (*Barennes et al., 2020*; *Liu et al., 2016*; *Figure 5c–d*, *Figure 6*).

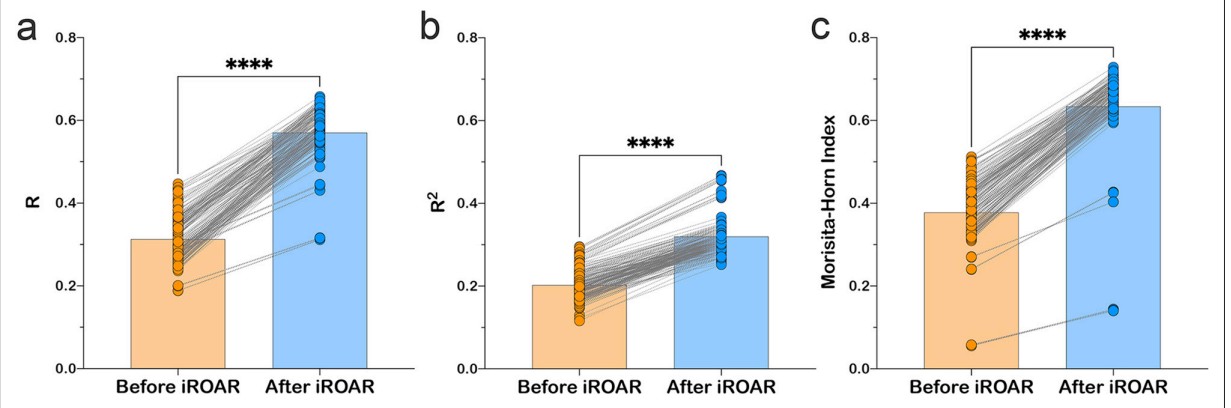

**Figure 6.** Effect of iROAR (immune Repertoire Over Amplification Removal)-based PCR bias correction in MPlex repertoire on similarity with low-biased Rapid Amplification of cDNA Ends (RACE)-based repertoire obtained from the same RNA sample. (**a**) Pearson correlation coefficient; (**b**) $R^2$ measure; (**c**) Morisita-Horn similarity index. **** $p < 0.0001$ (two-tailed Wilcoxon matched-pairs signed rank test, CI = 0.95). Dataset: PRJNA548335 three different RACE (RACE-2, RACE-3, and RACE-4 in six replicates each) protocols vs. RNA-based MPlex (Multiplex-3) protocol for Donor1 and Donor2 (100 ng RNA input): total 36 points; PRJNA309577 (one RACE protocol vs. one MPlex protocol) for Donors S01 (four MPlex replicates vs. two RACE replicates), S02 (two MPlex replicates vs. four RACE replicates), and donor S03 (one MPlex replicate vs. one RACE replicate): total 17 points.

The online version of this article includes the following source data for figure 6:

**Source data 1.** XLSX table.

As a result of amplification bias correction, OAR-index for multiplex-based repertoire decreased 1.5-fold average. At the exact time, the correlation of clonal frequencies obtained with RACE and multiplex significantly increased (Pearson correlation measure and $R^2$ value increased 1.5-fold average each) with a significant rise of repertoires similarity (Morisita-Horn index increased 1.7-fold average; *Figure 6*). Importantly, amplification bias decreased in both out-of-frame and in-frame clone sets, although normalization coefficients were calculated using out-of-frame ones only.

## Comparison of iROAR and spike-in-based approach for amplification bias detection

Biological spike-in is considered a classical technique for multiplex PCR bias evaluation. Several options for this technique including synthetic repertoire (*Carlson et al., 2013*; *Wu et al., 2020*), lymphoid cell lines DNA mix, and DNA from human blood, tonsil, and thymus (*Kallemeijn et al., 2018*; *Knecht et al., 2019*) were established to measure V- and J-segment specific primers performance during TCR/BCR rearrangements amplification in multiplex PCR. In this study, we compared iROAR-based amplification bias evaluation with a spike-in-based approach. Similarly to *Kallemeijn et al., 2018*; *Knecht et al., 2019*, we were using natural thymic cell-derived spike-ins rather than synthetic ones. Human CD8 T-cells derived DNA was used as a target input for the libraries' preparation. TRA rearrangements library of thymocytes were used as a source of spike-ins. Two different random mixes of TRAV- and TRAJ-specific primers (0.18–4.7 µM each) were used for multiplex PCR amplification of target DNA with spike-in added. Each test library was prepared in two replicas (four test libraries total). The obtained libraries were sequenced with an average coverage of 9.88 reads per clonotype

**Table 1.** The number of spike-in and target clonotypes in test TRA libraries.

| Sample | Spike-in clonotypes | | | Target clonotypes | | |
|---|---|---|---|---|---|---|
| | Number | Read count | Coverage | Number | Read count | Coverage |
| Primer mix 1 Replica 1 | 3571 | 30,474 | 8.53 | 39,911 | 348,792 | 8.74 |
| Primer mix 1 Replica 2 | 2698 | 19,420 | 7.20 | 35,818 | 303,494 | 8.47 |
| Primer mix 2 Replica 1 | 3439 | 34,717 | 10.10 | 40,209 | 425,508 | 10.58 |
| Primer mix 2 Replica 2 | 2298 | 24,823 | 10.80 | 33,406 | 383,615 | 11.48 |

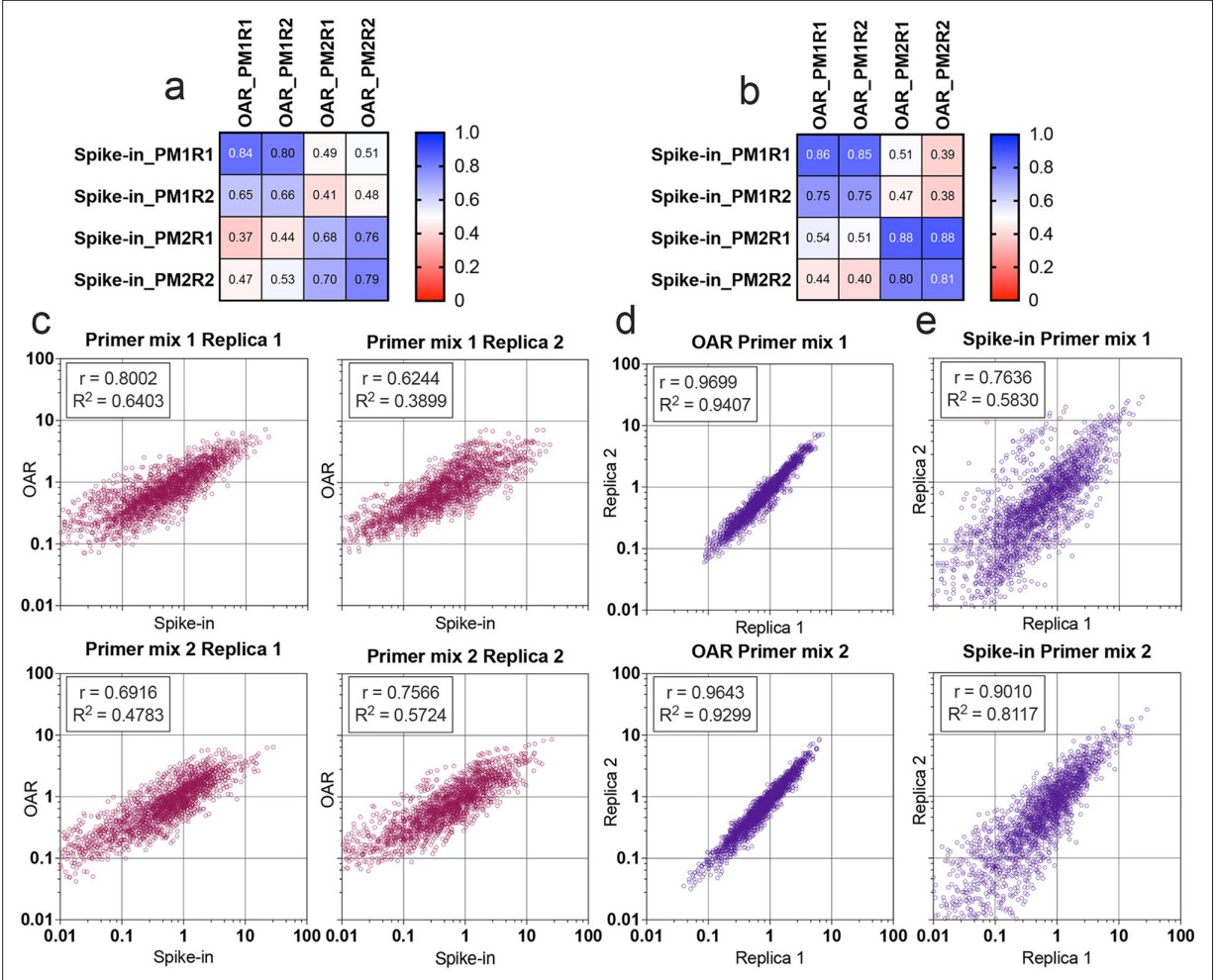

**Figure 7.** Comparison of Over Amplification Rate (OAR)-based and biological spike-in-based approaches for multiplex PCR bias detection. Pearson's correlation coefficient for V-segments bias measure (**a**) and J-segments bias measure (**b**). Column and row titles: PM = Primer mix, $R$=replica. (**c**) Correlation of VJ combination bias calculated by iROAR (immune Repertoire Over Amplification Removal) and biological spike-ins. (**d**) Reproducibility of iROAR-based VJ combination bias detection. (**e**) Reproducibility of spike-in-based VJ combination bias detection. Data: PRJNA825832.

The online version of this article includes the following source data for figure 7:

**Source data 1.** XLSX table.

and contained 35,818–40,209 target and 2298–3571 spike-in clonotypes after pseudogenes removal (**Table 1**).

Multiplex PCR bias of each separate V- and J-gene was calculated using both iROAR and biological spike-in approaches demonstrating the high-correlation level (**Figure 7a and b**) for the matched OAR/spike-in pairs (Pearson's $r$=0.78 average) in contrast to mismatched ones (Pearson's $r$=0.46 average). VJ combination bias for both approaches was calculated by multiplying V- and J-segment biases and compared using correlation analysis (**Figure 7c–e**). iROAR and spike-in detected VJ biases showed a strong positive correlation (Pearson's $r$=0.7182 average) for all four test TRA libraries (**Figure 7c**). Based on replicas comparison, the reproducibility of iROAR detected VJ bias was higher than one detected using spike-in control (**Figure 7d and e**).

## Impact of iROAR on a similarity of repertoires prepared by different multiplex PCR systems

To further test the iROAR approach's ability to raise the uniformity of repertoires by reducing multiplex PCR-specific bias, we analyzed changes in the similarity of repertoires prepared for the same individual but using different multiplex methods. For this purpose, we compared OARs, V/J, and

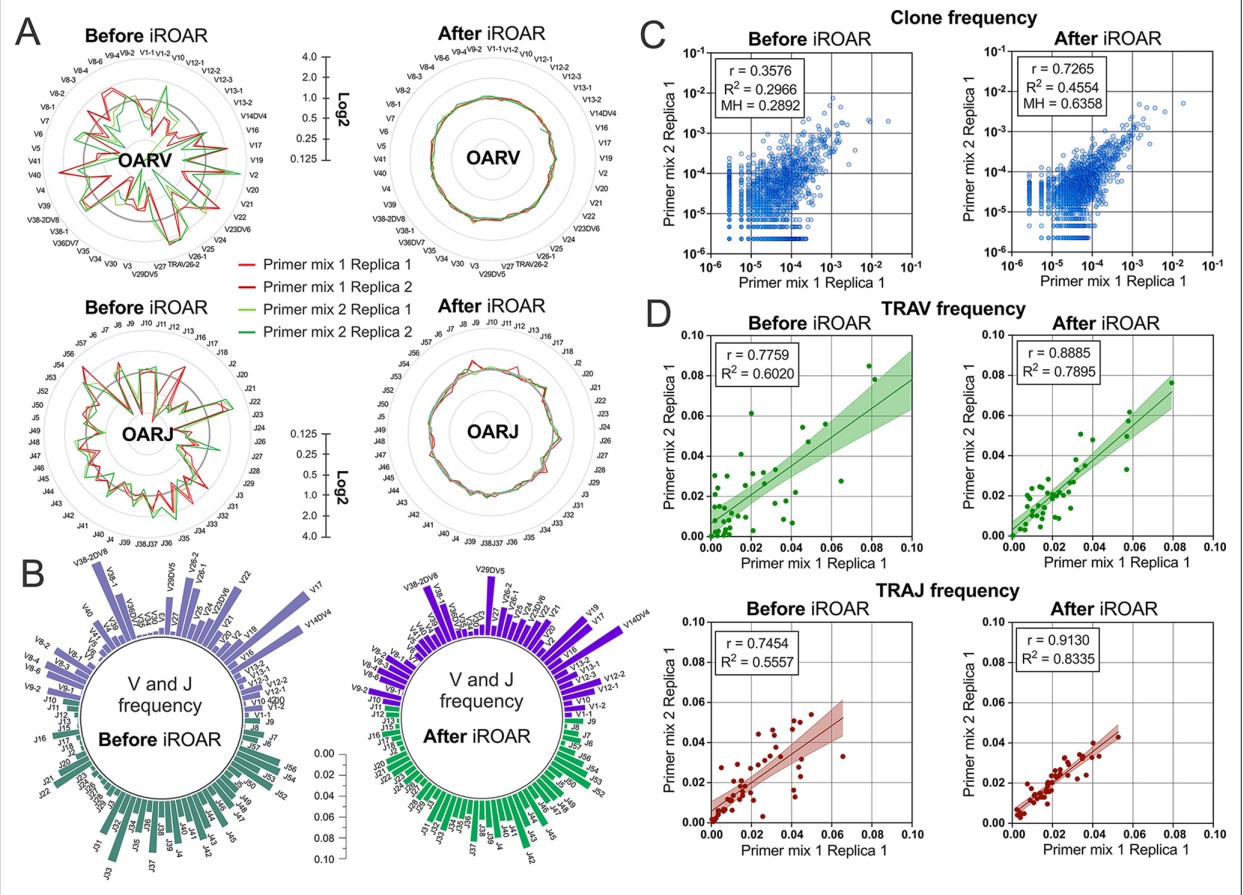

**Figure 8.** Convergence of Over Amplification Rate (OAR), clonotype, and V/J frequencies between two TRA repertoire before and after iROAR (immune Repertoire Over Amplification Removal) based bias correction. (**a**) OAR values changes in four test TRA libraries after PCR bias correction using iROAR. (**b**) TRAV and TRAJ frequency changes after PCR bias correction using iROAR (Sample: Primer mix 1 Replica 1). (**c**) Correlation of clonal frequencies of two different types of test TRA repertoires before and after iROAR-based PCR bias correction. (**d**) Correlation of V- and J-gene frequencies of test TRA repertoires before and after iROAR-based PCR bias correction.

The online version of this article includes the following source data for figure 8:

**Source data 1.** XLSX table.

clonotype frequencies before and after bias correction using iROAR in test TRA libraries prepared with Primer mix 1 and Primer mix 2 (after spike-in removal). As a result of iROAR-based bias correction, the difference between OARs for these two library types significantly decreases, and OARs themselves approach a value of one. By default, iROAR does not affect the diversity of repertoires and does not remove any clonotypes. Meanwhile, V and J frequencies are subject to substantial changes (*Figure 8b*) depending on the initial bias level. These changes occur in both biased repertoires (Primer mix 1 and Primer mix 2) and lead to an increase its convergence (*Figure 8d*). Herewith, $R^2$ measure increased 1.31-fold and 1.5-fold for V- and J-gene frequencies, respectively. Moreover, bias correction using iROAR also increases similarities of clone frequencies (*Figure 8c*). In this case, both the Morisita-Horn index and Pearson correlation coefficient increase twofold and $R^2$ measure increases 1.5-fold.

It is important to note that OARs calculation and bias correction for each of the analyzed test TRA repertoires was performed entirely independently without the involvement of any common normalization coefficients or spike-in controls. Therefore, each repertoire contains enough information to correct it adequately, increasing the consistencies of interrogated repertoires obtained even by different multiplex PCR protocols.

All observed results can be considered evidence of the actual capacity of iROAR approach to accurately detect and reduce multiplex-specific quantitative bias in adaptive immune receptor repertoires.

## Discussion

Even a small difference in amplification efficiencies can lead to a massive bias after multiple amplification cycles due to the exponential nature of PCR. Thus, most of the existing immune repertoire library preparation methods are subjected to amplification bias. The effect of distinct PCR bias-generating factors can be reduced experimentally by varying reaction mixture content and introducing special protocols (UMI, crafty primer structures, and spike-in controls). However, the criteria for estimating and removing the residual bias after applying these optimization approaches are lacking. Here, we close this gap by introducing the OAR value and OAR-index, which score PCR bias for both V- and J-genes separately (OAR values) and the whole repertoire dataset (OAR-index). Based on OAR values, we developed the first fully computational approach to decipher and correct amplification bias in adaptive immune receptor repertoires produced by one- or two-side multiplex PCR-based methods, using RNA or DNA as a template. Due to the inability to use UMI-based correction for DNA-based multiplex, the developed approach is the only currently available technique allowing direct measuring and correcting PCR bias in such repertoires without additional experiments.

In contrast to cell-line mix spike-in (*Knecht et al., 2019*) or synthetic repertoire-based (*Carlson et al., 2013*; *Wu et al., 2020*) PCR bias correction, the proposed approach operates with hundreds and even thousands of natural calibrators (out-of-frame clones) for each V-J gene pair. It makes this method potentially more reliable due to the ability to minimize the impact of CDR3 structure on PCR bias calculation since out-of-frame captures significantly higher CDR3 diversity than biological spike-ins. Moreover, similarly to a previously described method (*Carlson et al., 2013*), the OAR-based approach can also be used for primer efficacy evaluation to optimize their structures and concentrations, which in turn will straighten the coverage of various V- and J-genes and minimize the number of experimentally lost clones. Being fully computational, the developed PCR bias correction algorithm can be easily implemented in any TCR/BCR repertoire analysis pipeline, noticeably improving the quantitative parts of the analysis. Even though it's not possible to fully substitute the low-biased RACE methods, iROAR is capable to make multiplex-PCR-based repertoires more consistent with RACE-based ones. Therefore, the developed approach can provide the opportunity to compare the immune repertoire datasets generated using different library preparation methods.

## Methods

### Raw-data processing and immune repertoires reconstruction

All sequencing data used in this study represent human TCR and BCR repertoires. The repertoires (see *Supplementary file 1*) were reconstructed from fastq data using MiXCR v2 software (*Bolotin et al., 2017*; *Bolotin et al., 2015*) after primers and adapters trimming using FASTP software (*Chen et al., 2018*). All obtained repertoires were converted to VDJTOOLS (*Shugay et al., 2015*) format for unification. Erroneous clones generated by single nucleotide substitution were removed from the repertoire using the 'Correct' function from the VDJTOOLS software package. Erroneous clones generated by single nucleotide indels were removed from repertoires using the 'Filter' function from developed iROAR software. V and J pseudogenes were removed from repertoires using the 'FilterBySegment' function of VDJTOOLS.

### TRA repertoires preparation

The peripheral blood was collected from a healthy volunteer from the article's co-authors with informed consent in a certified clinical lab. PBMC was separated from whole blood using the Ficoll-Paque approach. CD8+ T cells were isolated using Dynabeads CD8 Positive Isolation Kit (Invitrogen). DNA for library preparation was extracted from CD8+ T cells using FlexiGene DNA Kit (Qiagen). 150 ng aliquots of obtained DNA were used as input to prepare each out of four TRA libraries. Each DNA aliquot was premixed with 0.1 pg of serial diluted low-biased TRA library (prepared using MiLaboratories Human TCR kit) of thymic cells (spike-in matrix) as biological spike-ins. Two pools of previously designed (*Komkov et al., 2020*) TRAV- and TRAJ-specific primers (MiLaboratories LLC) with randomly selected concentrations (0.18–4.7 μM each) were generated to produce two types of TRA libraries with different quantitative bias status simulating libraries produced by different multiplex PCR methods. Library preparation was performed according to the protocol from *Komkov et al., 2020*. Both types of TRA libraries were prepared in two replicas and sequenced along with a spike-in

matrix library on the MiSeq Illumina instrument (SE 150 nt) with moderate coverage 480,000 reads per library.

## Biological spike-in detection and analysis

TRA repertoires were extracted from FASTQ files using MIXCR software. All obtained MIXCR output files were converted to VDJTOOLS format as described above. Extraction of the spike-in sequences and spike-in-free repertoires from sequenced libraries was performed using the VDJTOOLS function 'ApplySampleAsFilter' and the sequenced spike-in library as a filter. Spike-in-based amplification bias was calculated as the quotient of V- and J-frequency in spike-ins extracted from target libraries and corresponding V- and J-frequency in the spike-in matrix, which was not subjected to multiplex amplification. OARs for obtained TRA libraries were calculated using iROAR software and spike-in-free repertoires as input. VJ bias values were calculated by multiplying V- to J-segment-specific biases. Correlation analysis of iROAR and spike-in VJ bias values was performed using GraphPad Prism9 software.

## Step-by-step pipeline for the OAR evaluation used in this study

1. Single nucleotide error correction in read1/read2 intersected sequences and Illumina adapters removal (optional): fastp -c -i input_R1.fastq.gz -I input_R2.fastq.gz -o fp.input_R1.fastq.gz -O fp.input_R2.fastq.gz.
2. Raw reads alignment (essential):
   a. For TCR beta chains mixcr align -c TRB fp.input_R1.fastq.gz fp.input_R2.fastq.gz output1.vdjca.
   b. For TCR beta chains mixcr align -c TRA fp.input_R1.fastq.gz fp.input_R2.fastq.gz output1.vdjca.
3. Clonotypes assemble (essential): mixcr assemble output1.vdjca output2.clns
4. TCR repertoire export in a human-readable format (essential): mixcr export Clones output2.clns clones.txt.
5. Convert repertoire into VDJtools format (essential):java -jar vdjtools.jar Convert -S mixcr clones.txt vdjtools.
6. Artificial diversity removal by single nucleotide substitutions correction (optional):java -jar vdjtools.jar Correct vdjtools.clones.txt correct.
7. Pseudogenes removal (optional):
   a. For TCR beta chains.java -jar vdjtools.jar FilterBySegment –-j-segment TRBJ2-2P --v-segment TRBV1,TRBV12-1,TRBV12-2,TRBV17,TRBV21-1,TRBV22-1,TRBV23-1,TRBV26,TRBV5-2,TRBV5-3,TRBV5-7,TRBV6-7,TRBV7-1,TRBV7-5,TRBV8-1,TRBV8-2 --negative correct.vdjtools.clones.txt filter
   b. For TCR alpha chainsjava -jar vdjtools.jar FilterBySegment –-j-segment TRAJ1,TRAJ19,TRAJ2,TRAJ25,TRAJ51,TRAJ55,TRAJ58,TRAJ59,TRAJ60,TRAJ61 --v-segment TRAV11,TRAV11-1,TRAV14-1,TRAV15,TRAV28,TRAV31,TRAV32,TRAV33,TRAV37,TRAV46,TRAV8-5,TRAV8-6-1,TRAV8-7 --negative correct.vdjtools.clones.txt filter
8. Artificial diversity removal by single nucleotide indels correction (optional): iroar Filter -se 0.01 filter.correct.vdjtools.clones.txt filter2.txt
9. OARs calculation and quantitative bias correction (essential): iroar Count -min_outframe 15 r -z 1 -iter 1 -mt 1 input_folder output_folder (input_folder must contain filter2.txt file).

## OAR evaluation and statistical analysis

For the OAR and OAR-index calculation and amplification bias removal, we used the command-line-based iROAR software designed in this study and freely available for non-profit use at GitHub (https://github.com/smiranast/iROAR; *Komkov, 2023*; copy archived at swh:1:rev:2362c4f41d40519154e1c-2dc6ce7af619f15fb4b). For the OAR comparison between 5'-RACE, one-side, and two-side multiplex PCRs, an equal number of out-of-frame clones (50,000) was randomly selected from TCR repertoires of 15 healthy individuals (for each approach). Average population V- and J-gene frequencies (unweighted) were calculated based on out-of-frame clones from 105 TRB repertoires obtained by two methods: 5'-RACE (95 repertoires) and single-cell TCR profiling (10× Genomics; 10 repertoires; *Supplementary file 1*) using the 'CalcSegmentUsage' function with '-u' parameter of VDJTOOLS. All statistical tests were performed using Prism9 GraphPad software (https://www.graphpad.com/).

## iROAR software requirement

Recommended system configuration for iROAR running: Linux or MacOS, 2 CPU, 8 GB RAM, programming language: python = 3.7.3, required Python packages: matplotlib = 3.0.3, numpy = 1.16.2, pandas = 0.24.2, and requests = 2.21.0. Starting iROAR package includes list of average populational frequencies with SDs of TRB and TRA V- and J-genes related to the European population. iROAR run command: iroar Count (optional parameters) <input> . Recommended parameters for most tasks: -min_outframe 15r -z 1 -iter 1 -mt 1. Full list of available parameters is deposited in project directory at github (https://github.com/smiranast/iROAR).

## Data access

All analyzed datasets were downloaded from open-source databases: NCBI SRA (https://www.ncbi.nlm.nih.gov/sra), ENA (https://www.ebi.ac.uk/ena), and Zenodo project (https://zenodo.org/). A complete list of web links and accession numbers is summarized in *Supplementary file 1*. TRA repertoire dataset generated in this study for iROAR validation is available under access number PRJNA825832.

The iROAR software and its documentation are available at the link: https://github.com/smiranast/iROAR. The additional software used in this study is available in the GitHub repository: MiXCR v2 (*Bolotin et al., 2015*) (https://github.com/milaboratory/mixcr; *MiLaboratories, 2023*), VDJTOOLS (*Shugay et al., 2015*) ( https://github.com/mikessh/vdjtools; *Shugay, 2022*), and FASTP (*Chen et al., 2018*) (https://github.com/OpenGene/fastp; *OpenGene - Open Source Genomics Toolbox, 2022*).

## Acknowledgements

We thank Grigory Armeev and Valery Novoseletsky for their help with data storage and processing. This work was supported by Russian Science Foundation (grant 20-75-10091 to A.K.), the analysis of TRA MPlex data was supported by Russian Foundation for Basic Research (grant 20-015-00462 to A.K.).

## Additional information

### Funding

| Funder | Grant reference number | Author |
|---|---|---|
| Russian Science Foundation | 20-75-10091 | Alexander Komkov |
| Russian Foundation for Basic Research | 20-015-00462 | Alexander Komkov |

The funders had no role in study design, data collection and interpretation, or the decision to submit the work for publication.

### Author contributions

Anastasia O Smirnova, Data curation, Software, Formal analysis, Investigation; Anna M Miroshnichenkova, Validation, Writing – review and editing; Yulia V Olshanskaya, Michael A Maschan, Yuri B Lebedev, Dmitriy M Chudakov, Resources, Supervision, Writing – review and editing; Ilgar Z Mamedov, Resources, Supervision, Funding acquisition, Writing – original draft, Project administration, Writing – review and editing; Alexander Komkov, Conceptualization, Formal analysis, Supervision, Funding acquisition, Investigation, Methodology, Writing – original draft, Project administration, Writing – review and editing

### Author ORCIDs

Yuri B Lebedev ⓘ http://orcid.org/0000-0003-4554-4733
Dmitriy M Chudakov ⓘ http://orcid.org/0000-0003-0430-790X
Alexander Komkov ⓘ http://orcid.org/0000-0001-9113-698X

Decision letter and Author response
Decision letter https://doi.org/10.7554/eLife.69157.sa1

## Additional files

### Supplementary files
• Supplementary file 1. XLSX table. Dataset accession numbers and references to the original studies generated it.

• Supplementary file 2. XLSX table. Comparison of iROAR (immune Repertoire Over Amplification Removal) algorithm with the existing approaches for PCR bias removal in human adaptive immune receptor repertoires.

• Transparent reporting form

### Data availability

Sequencing data have been deposited in SRA under accession code PRJNA825832. All other sequencing data analyzed during this study are previously published and fully available under links or access numbers included in the manuscript and supporting files.

The following dataset was generated:

| Author(s) | Year | Dataset title | Dataset URL | Database and Identifier |
|---|---|---|---|---|
| Smirnova A | 2022 | TRA repertoire | https://www.ncbi.nlm.nih.gov/bioproject/PRJNA825832/ | NCBI BioProject, PRJNA825832 |

The following previously published datasets were used:

| Author(s) | Year | Dataset title | Dataset URL | Database and Identifier |
|---|---|---|---|---|
| Warren RL, Freeman JD, Zeng T, Choe G, Munro S | 2011 | Exhaustive T-cell repertoire sequencing of human peripheral blood samples reveals signatures of antigen selection and a directly measured repertoire size of at least 1 million clonotypes | https://www.ncbi.nlm.nih.gov/sra/?term=SRA020989 | NCBI Sequence Read Archive, SRA020989 |
| Zvyagin IV | 2014 | Homo sapiens T-cell repertoire - MZ twins | https://www.ncbi.nlm.nih.gov/bioproject/?term=PRJNA214848 | NCBI BioProject, PRJNA214848 |
| Rosati E | 2020 | TCR repertoire in IBD twins | https://www.ncbi.nlm.nih.gov/bioproject/?term=PRJEB27352 | NCBI BioProject, PRJEB27352 |
| de Greef PC | 2020 | TCR repertoire sequencing of T cell subsets from healthy individuals | https://www.ncbi.nlm.nih.gov/bioproject/?term=PRJNA390125 | NCBI BioProject, PRJNA390125 |
| Pogorelyy MV | 2018 | Precise tracking of vaccine-responding T-cell clones reveals convergent and personalized response in identical twins | https://www.ncbi.nlm.nih.gov/bioproject/?term=PRJNA493983 | NCBI BioProject, PRJNA493983 |
| Turchaninova et al | 2016 | Protocol for full length profiling of IG repertoires | https://www.ncbi.nlm.nih.gov/bioproject/?term=PRJNA297771 | NCBI BioProject, PRJNA297771 |
| Minervina et al | 2020 | Comprehensive analysis of antiviral adaptive immunity formation and reactivation down to single cell level | https://www.ncbi.nlm.nih.gov/bioproject/?term=PRJNA577794 | NCBI BioProject, PRJNA577794 |

*Continued on next page*

*Continued*

| Author(s) | Year | Dataset title | Dataset URL | Database and Identifier |
|---|---|---|---|---|
| Simon et al | 2018 | Sequencing the Peripheral Blood B and T cell Repertoire - Quantifying robustness and limitations | https://www.ncbi.nlm.nih.gov/bioproject/?term=PRJNA494572 | NCBI BioProject, PRJNA494572 |
| Pan et al | 2019 | Identification of drug-specific public TCR driving severe cutaneous adverse reactions | https://www.ncbi.nlm.nih.gov/bioproject/?term=PRJNA550004 | NCBI BioProject, PRJNA550004 |
| Ma et al | 2018 | Homo sapiens Raw sequence reads | https://www.ncbi.nlm.nih.gov/bioproject/?term=PRJNA427746 | NCBI BioProject, PRJNA427746 |
| Truong et al | 2019 | TCR diversity and clonality of human CD4+ memory T cells | https://www.ncbi.nlm.nih.gov/bioproject/?term=PRJEB31283 | NCBI BioProject, PRJEB31283 |
| Simnica et al | 2019 | Immunoaging | https://www.ncbi.nlm.nih.gov/bioproject/?term=PRJEB33490 | NCBI BioProject, PRJEB33490 |
| Weinberger J, Jimenez-Heredia R, Schaller S, Suessner S, Sunzenauer J, Reindl-Schwaighofer R, Weiss R, Winkler S, Gabriel C, Danzer M, Oberbauer R | 2015 | Immune repertoire profiling reveals that clonally expanded B and T cells infiltrating diseased human kidneys can also be tracked in the blood | https://doi.org/10.5281/zenodo.27483 | Zenodo, 10.5281/zenodo.27483 |
| Tanno et al | 2019 | Paired TCR alpha:TCR beta sequencing at the single-cell level | https://www.ncbi.nlm.nih.gov/bioproject/?term=PRJNA593622 | NCBI BioProject, PRJNA593622 |
| Liu et al | 2016 | TRB and IGH are captured from peripheral blood using Multiplex PCR and 5'RACE | https://www.ncbi.nlm.nih.gov/bioproject/?term=PRJNA309577 | NCBI BioProject, PRJNA309577 |
| Barennes T | 2020 | Benchmarking of T cell receptor repertoire profiling methods reveals large systematic biases | https://www.ncbi.nlm.nih.gov/bioproject/?term=PRJNA548335 | NCBI BioProject, PRJNA548335 |

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
