## [Editor Report]

This paper describes a newly developed, publicly available algorithm (iROAR) that was tested on pre-exisiting datasets and is of interest to T and B cell immunologists who perform repertoire analysis via multiplex PCR based techniques. iROAR utilises naturally occurring non-functional sequences to improve and partially correct the amplification bias inherent in multiplex PCR based sequencing technologies.

---

## [Decision Letter]

**Decision letter after peer review:**

Thank you for submitting your article "The use of non-functional clonotypes as a natural spike-in for multiplex PCR bias correction in immune receptor repertoire profiling" for consideration by *eLife*. Your article has been reviewed by 2 peer reviewers, and the evaluation has been overseen by a Reviewing Editor and Tadatsugu Taniguchi as the Senior Editor. The following individuals involved in review of your submission have agreed to reveal their identity: Lindsay Cowell (Reviewer #2).

Essential revisions:

The reviewers are enthusiastic about this manuscript. They have identified a number of issues that require clarification and are generally thought to be important to reach the broad immunology audience.

*Reviewer #1 (Recommendations for the authors):*

The conclusions are mostly well supported. However, there are some concerns that should be addressed before this paper is accepted for publication.

The authors have kindly provided the iROAR software free for non-profit use on Github. As this entire study was performed to validate and share this software to other scientists, I would suggest that the iROAR documentation be improved such that it can be used by a wide audience. More detailed instructions for use and perhaps even a step by step example on how to replicate the results in this study included in the documentation would be very helpful.

The authors utilised pre-existing datasets that performed both multiplex PCR (high bias) and 5'-RACE (low bias) on the same sample to test the functionality of iROAR. The iROAR algorithm improved the PCR bias in repertoires that were obtained by multiplex PCR which better correlated with the same repertoire determined by 5'-RACE. It is important to note that although the iROAR algorithm improved PCR bias, it is by no means perfect and cannot substitute for a low bias approach. Even for a 5'-RACE repertoire in which artificial bias was introduced in silico, iROAR improved the correlation of the in silico biased 5'-RACE from R2 = 0.4052 to R2 = 0.6286 which is an improvement but certainly cannot substitute for a repertoire that was determined using a low bias approach.

The authors very nicely showed the extent of amplification bias when using multiplex PCR based technologies by plotting the OAR distributions in Figure 2A and Figure 2B. It would be very informative to plot the iROAR corrected OAR and OAR distributions as well in order to visualise the improvement in bias correction in the same way..

The authors state that an F-test was performed to compare OAR distributions in Figure 2A. Please state exactly which F-test was used to calculate statistical significance.

The authors state that they use a z-test to exclude outliers from the OAR calculation that may have resulted from large clonotypes introduced by PCR bias or naturally generated from clonal expansions. The authors should describe further how this affects analysis of repertoires in which large clonal expansions are expected such as during an antiviral immune response.

The authors nicely compared clonal frequency between iROAR corrected and 5'-RACE repertoires in Figure 4B-D. I think it would also be important to know how the iROAR algorithm would affect other measurements commonly used when analysing TCR/BCR repertoires. For example, does iROAR also affect measurements of diversity? This would be especially useful when comparing corrected repertoires vs. 5'-RACE.

As per *eLife* policy, "Regardless of whether authors use original data or are reusing data available from public repositories, they must provide program code, scripts for statistical packages, and other documentation sufficient to allow an informed researcher to precisely reproduce all published results." I would recommend that the authors very much improve the documentation provided in the Github repository for iROAR such that anyone can reproduce the published results before publication. Additionally, *eLife* suggests that the authors should license their code using an open source license.

Line 158: What do you mean by z-test? Can you explain under which circumstances you can decide to exclude abnormally large clonotypes? How has this threshold been calculated? Would this negatively affect measurements of OAR during an immune response such as viral infections where large clonotypes might be observed?

Line 167, 2.5 reads is a minimal sufficient sequencing coverage. Does the package provide this quality control check? This would be useful information that could be included in the manuscript as well as within iROAR documentation.

Line 203 Figure 4: Can the authors also generate the OAR distributions (a per Figure 2B) for these datasets before and after iROAR? Would be very useful as a comparison between 5' RACE and other multiplex PCR methods as well as to assess the efficacy of iROAR in correcting bias.

Line 219, How does the in silico introduced bias look with regards to the OAR? Can you generate the OAR distributions of in silico biased and unbiased samples using the same graphing method in Figure2B?

The Discussion section needs much more discussion on its applications as well as limitations. Ie., how does the software deal with instances where clonal expansions occur in cell clones that are also bearing a non-functional receptor? How does this affect the OAR calculation?

*Reviewer #2 (Recommendations for the authors):*

Enthusiasm is dampened by the fact that the proposed method is not directly compared to the gold standard of biological spike-ins. The results would be stronger if the authors could use data generated with a biological spike-in and compare correction using data from the spike-in versus their proposed algorithm. If such a data set is publicly available or can be obtained by request, this should be done. If such a data set cannot be obtained, then the Discussion section should directly address the fact that this gold standard validation remains to be done.

Figure 2:

– More interpretation of the results is needed in the Discussion. For example, Figure 2a appears to show that for VMPlex and VJMPlex, most genes suffer from under- rather than over-amplification. This seems unexpected. I would have expected "zero-sum-game" behavior. Also, it is surprising that VJMPlex shows less overall bias than VMPlex. Does this make sense? Why or why not? I think a plot like Figure 1 is needed for VMPlex and VJMPlex to ensure that the base assumption underlying OAR calculation holds in the same way and to the same degree across the three technologies.

– Regarding the input data: "for each method type 15 PBMC TCR repertoires were chosen randomly from" and this is followed by a list of 11 study/project identifiers. More information is needed. Fifteen seems like a small number, but it depends on the details. How do the 11 studies distribute across the three methods? How many studies were included in each method? Were 15 repertoires total used for each method? Or 15 from each study? Did the different studies within a method use the same primer set? The same depth coverage?

– How much within-study between-repertoire variability is there for a single V or J gene? And how much variability is there across studies? It seems important to understand this basic behavior of the metric before pooling into a single figure, as the pooling can average out important behavior.

Figure 3: More interpretation of the results is needed.

– In Figure 3a, the range of values is greater for TRBV and IGHV. Could this be attributed to a larger number of genes or primers, or to more sequence variability (and presumably therefore greater variability in primer hybridization efficiency)?

– In Figure 3b, while there may be no difference in the mean or median OAR value across cell types, there does appear to be a difference in the range, with PBMCs and THs showing much less variability than the other cell types. What could explain this? Is this an artifact of the way samples were pooled for the figure? Are both donors represented in all six cell types? Or does this point to some interesting biology? If the latter, then exploration of this is understandably beyond the scope of this paper, but it should be mentioned and possible explanations should be put forward.

Lines 197 – "the procedure can be recursively repeated with a modified normalization coefficient defined as described coefficient raised to the power of a number in the range from 0 to 1.": How is the value of this number determined? This sounds like an optimization procedure, in which case much more information is needed. In particular: what algorithm was used? what is the objective function? What are the stopping criteria? In which of the presented results was this procedure applied?

Line 87 "constant during a lifetime": this needs to be substantiated with either a reference or data. The authors show relevant data in Supplementary Figure 1, but there is no description of how the data shown in the figure were derived. For example: are all data points from a single study? do all data points used in a single regression (i.e., all those corresponding to the same gene segment) derive from the same patient? do all data in the figure derive from one single patient? what sequencing protocol was used? Has the proposed correction been applied? The strength of the claim should reflect the strength of the data. Note: I don't think the truth of this claim is a requirement for the proposed method to be valid. So if the data are such that the statement cannot be generalized to "all" repertoires, I don't see this as a problem.

Line 89 "reproducible": reproducible in what way? Over multiple aliquots of a sample? Over multiple samples for the same person? Over multiple people?

Figure 1: Is the same stability observed for the sequencing protocols with even more amplification bias (VMPlex and VJMPlex)?

Lines 102 – 109: this paragraph is not clearly written, but it is the heart of the manuscript. The paragraph reads as if only the two terms with summations are based on out-of-frame rearrangements. I assume all four terms are based on out-of-frame rearrangements? Because otherwise the equation does not make sense …. Also, I recommend removing "a percentage of" in line 105, as this reads as if there is a percentage in the numerator of the numerator (i.e., the RC(Vi) term). Finally, in line 107, I assume you mean PCR amplification instead of clonal expansion?

Figure 2: More information about the data is needed. Specific details are below.

Lines 149 – 153 starting with "the average population frequencies": It isn't clear what is meant by "average population frequencies", and it isn't clear what the rest of the paragraph implies about calculations described in the paper and results displayed in the figures. It sounds as if different repertoires may have been subjected to different calculations.

Lines 157 – 158: The last sentence of this paragraph suggests that V- or J-genes with large or small relative frequencies may have been excluded. More details are needed. Were only large clones excluded? How were they identified for exclusion? By what method and what exclusion threshold was used? How many were excluded from each repertoire? If more than "a few", then before and after data need to be shown for some repertoires to show how this impacts the OAR distributions and averages.

Lines 167 – 169: how is "adequate" defined? Within the 10% error range discussed in the context of Supplemental Figure 3?

Lines 257 – 262: These sentences suggest that the proposed approach is more reliable than biological spike-ins. This should be substantiated before being claimed. The authors state that this is because "the impact of CDR3 structure" is minimized. Why is this? Because the number of out-of-frame rearrangements is much much larger than the number of cell-line or synthetic spike-ins and so captures more CDR3 diversity? Or some other reason? More explanation of why this is being suggested should be given and the wording should make clear that this is an untested hypothesis.

---

## [Author Response]

Reviewer #1 (Recommendations for the authors):The conclusions are mostly well supported. However, there are some concerns that should be addressed before this paper is accepted for publication.The authors have kindly provided the iROAR software free for non-profit use on Github. As this entire study was performed to validate and share this software to other scientists, I would suggest that the iROAR documentation be improved such that it can be used by a wide audience. More detailed instructions for use and perhaps even a step by step example on how to replicate the results in this study included in the documentation would be very helpful.

We edited the documentation at GitHub and provided step-by-step instructions for the general usage in the Methods section “Step-by-step pipeline for the OAR evaluation used in this study”:

Single nucleotide error correction in read1/read2 intersected sequences and Illumina adapters removal (optional):

fastp -c -i input_R1.fastq.gz -I input_R2.fastq.gz -o fp.input_R1.fastq.gz -O fp.input_R2.fastq.gz

Raw reads alignment (essential):

For TCR β chains

mixcr align -c TRB fp.input_R1.fastq.gz fp.input_R2.fastq.gz output1.vdjca

For TCR β chains

mixcr align -c TRA fp.input_R1.fastq.gz fp.input_R2.fastq.gz output1.vdjca

Clonotypes assemble (essential):

mixcr assemble output1.vdjca output2.clns

TCR repertoire export in a human-readable format (essential):

mixcr exportClones output2.clns clones.txt

Convert repertoire into VDJtools format (essential):

java -jar vdjtools.jar Convert -S mixcr clones.txt vdjtools

Artificial diversity removal by single nucleotide substitutions correction (optional):

java -jar vdjtools.jar Correct vdjtools.clones.txt correct

Pseudogenes removal (optional):

For TCR β chains

java -jar vdjtools.jar FilterBySegment –-j-segment TRBJ2-2P --v-segment TRBV1,TRBV12-1,TRBV12-2,TRBV17,TRBV21-1,TRBV22-1,TRBV23-1,TRBV26,TRBV5-2,TRBV5-3,TRBV5-7,TRBV6-7,TRBV7-1,TRBV7-5,TRBV8-1,TRBV8-2 –-negative correct.vdjtools.clones.txt filter

For TCR α chains

java -jar vdjtools.jar FilterBySegment -–j-segment TRAJ1,TRAJ19,TRAJ2,TRAJ25,TRAJ51,TRAJ55,TRAJ58,TRAJ59,TRAJ60,TRAJ61 --v-segment TRAV11,TRAV11-1,TRAV14-1,TRAV15,TRAV28,TRAV31,TRAV32,TRAV33,TRAV37,TRAV46,TRAV8-5,TRAV8-6-1,TRAV8-7 –-negative correct.vdjtools.clones.txt filter

Artificial diversity removal by single nucleotide indels correction (optional):

iroar Filter -se 0.01 filter.correct.vdjtools.clones.txt filter2.txt

OARs calculation and quantitative bias correction (essential):

iroar Count -min_outframe 15 -z 1 -mt 1 -iter 1 input_folder output_folder

(input_folder must contain filter2.txt file)The authors utilised pre-existing datasets that performed both multiplex PCR (high bias) and 5'-RACE (low bias) on the same sample to test the functionality of iROAR. The iROAR algorithm improved the PCR bias in repertoires that were obtained by multiplex PCR which better correlated with the same repertoire determined by 5'-RACE. It is important to note that although the iROAR algorithm improved PCR bias, it is by no means perfect and cannot substitute for a low bias approach. Even for a 5'-RACE repertoire in which artificial bias was introduced in silico, iROAR improved the correlation of the in silico biased 5'-RACE from R2 = 0.4052 to R2 = 0.6286 which is an improvement but certainly cannot substitute for a repertoire that was determined using a low bias approach.

We added a suggested sentence: “Even though it’s not possible to fully substitute the low-biased RACE methods, iROAR is capable to make multiplex-PCR-based repertoires more consistent with RACE based ones.”

The authors very nicely showed the extent of amplification bias when using multiplex PCR based technologies by plotting the OAR distributions in Figure 2A and Figure 2B. It would be very informative to plot the iROAR corrected OAR and OAR distributions as well in order to visualise the improvement in bias correction in the same way.

Done. Figures 5 and 8.

The authors state that an F-test was performed to compare OAR distributions in Figure 2A. Please state exactly which F-test was used to calculate statistical significance.

We used nonparametric *Levene's test*. We specified it in the Figure 2 legend.

The authors state that they use a z-test to exclude outliers from the OAR calculation that may have resulted from large clonotypes introduced by PCR bias or naturally generated from clonal expansions. The authors should describe further how this affects analysis of repertoires in which large clonal expansions are expected such as during an antiviral immune response.

It’s just a simple mathematical issue due to the formula which we used for OAR calculation. The dramatic error in OAR calculation can occur only in the case of enormous lymphoproliferation far beyond normal antiviral immune response.

We corrected the related statement in the manuscript:

Also, the balance of V- and J-genes frequencies can be disrupted by accidentally arisen abnormally large non-functional clonotypes either artificially amplified due to the extreme PCR bias or naturally generated in the course of abnormal clonal expansion in various lymphoproliferative disorders or stochastic spike in normal lymphocyte population. To reduce the impact of this anomaly on OAR value, the top clone of each V and J-gene containing subgroups must be excluded from OAR calculation. Since V- and J-specific bias affects all clones non-selectively, the remaining large part of clones after top clones exclusion is still representative for PCR bias calculation. As shown in Figure 4A, the exclusion of one top clonotype from OAR calculation for RACE-based TRB repertoire is enough to restore OAR calculation accuracy for TRBV2, TRBV5-6, TRBV7-9, TRBV11-3. The further top clones exclusion has no significant effect on OAR values.

The authors nicely compared clonal frequency between iROAR corrected and 5'-RACE repertoires in Figure 4B-D. I think it would also be important to know how the iROAR algorithm would affect other measurements commonly used when analysing TCR/BCR repertoires. For example, does iROAR also affect measurements of diversity? This would be especially useful when comparing corrected repertoires vs. 5'-RACE.

By default, iROAR does not affect diversity since it does not remove any clones. This statement was added to the manuscript.

As per eLife policy, "Regardless of whether authors use original data or are reusing data available from public repositories, they must provide program code, scripts for statistical packages, and other documentation sufficient to allow an informed researcher to precisely reproduce all published results." I would recommend that the authors very much improve the documentation provided in the Github repository for iROAR such that anyone can reproduce the published results before publication. Additionally, eLife suggests that the authors should license their code using an open source license.

We provided more detailed documentation of iROAR on GitHub and added step-by-step instructions in the Methods section. iROAR is totally free to the scientific community for non-profit usage.

Line 158: What do you mean by z-test? Can you explain under which circumstances you can decide to exclude abnormally large clonotypes? How has this threshold been calculated? Would this negatively affect measurements of OAR during an immune response such as viral infections where large clonotypes might be observed?

In the current version of iROAR we deleted z-test to improve calculation speed. However, we kept the option to discount the user-specified number of top non-functional clones (for each V and J gene) from the OAR calculation. It could significantly improve the accuracy of OAR calculation in case of strong clonal expansion, especially in lymphoproliferative disorders. To illustrate the impact of one top clone exclusion on OAR calculation improvement we added Figure 4a and the related text.

Line 167, 2.5 reads is a minimal sufficient sequencing coverage. Does the package provide this quality control check? This would be useful information that could be included in the manuscript as well as within iROAR documentation.

Done. This information is added to the iROAR report file.

Line 203 Figure 4: Can the authors also generate the OAR distributions (a per Figure 2B) for these datasets before and after iROAR? Would be very useful as a comparison between 5' RACE and other multiplex PCR methods as well as to assess the efficacy of iROAR in correcting bias.

Done. Figures 5 and 8.

Line 219, How does the in silico introduced bias look with regards to the OAR? Can you generate the OAR distributions of in silico biased and unbiased samples using the same graphing method in Figure2B?

Done. Figure 5.

The Discussion section needs much more discussion on its applications as well as limitations. Ie., how does the software deal with instances where clonal expansions occur in cell clones that are also bearing a non-functional receptor? How does this affect the OAR calculation?

To deal with this situation we recommend excluding the strongly expanded (top) non-functional clones from OAR calculation, as we mention above in responses as well as in the manuscript.

Reviewer #2 (Recommendations for the authors):Enthusiasm is dampened by the fact that the proposed method is not directly compared to the gold standard of biological spike-ins. The results would be stronger if the authors could use data generated with a biological spike-in and compare correction using data from the spike-in versus their proposed algorithm. If such a data set is publicly available or can be obtained by request, this should be done. If such a data set cannot be obtained, then the Discussion section should directly address the fact that this gold standard validation remains to be done.

During manuscript revision, we designed and performed a wet-lab experiment to directly compare the iROAR approach with biological spike-ins.

Figure 2:– More interpretation of the results is needed in the Discussion. For example, Figure 2a appears to show that for VMPlex and VJMPlex, most genes suffer from under- rather than over-amplification. This seems unexpected. I would have expected "zero-sum-game" behavior. Also, it is surprising that VJMPlex shows less overall bias than VMPlex. Does this make sense? Why or why not? I think a plot like Figure 1 is needed for VMPlex and VJMPlex to ensure that the base assumption underlying OAR calculation holds in the same way and to the same degree across the three technologies.

Thanks a lot for this observation. OAR coefficient is actually a relative measure. In the initial manuscript version, we used unnormalized OARs which can lead to misinterpretation of OAR values compared to other PCR bias measures. We corrected this inconvenience by using OAR normalized by average OAR value. This change can’t affect the bias removal process and at the same time will help to present bias coefficients in a more convenient way.

– Regarding the input data: "for each method type 15 PBMC TCR repertoires were chosen randomly from" and this is followed by a list of 11 study/project identifiers. More information is needed. Fifteen seems like a small number, but it depends on the details. How do the 11 studies distribute across the three methods? How many studies were included in each method? Were 15 repertoires total used for each method? Or 15 from each study? Did the different studies within a method use the same primer set? The same depth coverage?

Despite the number of published studies which used multiplex PCR-based methods for TCR profiling the number of publicly available raw read datasets is dramatically limited especially deep-sequenced ones. For this reason, the number of repertoires used for the analysis presented in Figure 2 was limited by available multiplex-based datasets (15 repertoires total for each method, i.e. 45 repertoires in sum, one repertoire for one individual). However, to address the concerns pointed out below we had to reduce this number to 6 repertoires per method. We corrected the figure legend according to the recommendation.

To equalize the sequencing coverage, we performed the down sampling as described in the Methods section.

– How much within-study between-repertoire variability is there for a single V or J gene? And how much variability is there across studies? It seems important to understand this basic behavior of the metric before pooling into a single figure, as the pooling can average out important behavior.

We corrected Figure 2 in order to show within-study between-repertoire variabilities.

Figure 3: More interpretation of the results is needed.– In Figure 3a, the range of values is greater for TRBV and IGHV. Could this be attributed to a larger number of genes or primers, or to more sequence variability (and presumably therefore greater variability in primer hybridization efficiency)?

Thanks for the questions. We truly appreciate such attention to even small details of our research. The variability of IGHV’s OARs is smaller compared to multiplex PCR but a little bit greater than TCR’s OARs. Probably, it’s a consequence of differences between clonal dynamics of particular subsets of Т-cell and B-cell. We agreed, that it’s interesting phenomenon, it’s indeed needed to be investigated separately. But we can only speculate about it for now.

Added to manuscript:

“Herewith, the variance of IGHV’s OARs compared TCRs’ and the variance of TCR subpopulations' OARs compared PBMCs’ is slightly higher. This phenomenon may be linked to well-known differences in clonal expansion intensities of B/T-cell subsets. But the proof of this hypothesis demands separate deep analysis which is beyond the main focus of this research.”

– In Figure 3b, while there may be no difference in the mean or median OAR value across cell types, there does appear to be a difference in the range, with PBMCs and THs showing much less variability than the other cell types. What could explain this? Is this an artifact of the way samples were pooled for the figure? Are both donors represented in all six cell types? Or does this point to some interesting biology? If the latter, then exploration of this is understandably beyond the scope of this paper, but it should be mentioned and possible explanations should be put forward.

Added to manuscript:

“Herewith, the variance of IGHV’s OARs compared TCRs’ and the variance of TCR subpopulations' OARs compared PBMCs’ is slightly higher. This phenomenon may be linked to well-known differences in clonal expansion intensities of B/T-cell subsets. But the proof of this hypothesis demands separate deep analysis which is beyond the main focus of this research.”

Lines 197 – "the procedure can be recursively repeated with a modified normalization coefficient defined as described coefficient raised to the power of a number in the range from 0 to 1.": How is the value of this number determined? This sounds like an optimization procedure, in which case much more information is needed. In particular: what algorithm was used? what is the objective function? What are the stopping criteria? In which of the presented results was this procedure applied?

The most quantitative bias can be corrected in the first iteration. (We added the corresponding sentence to the main text). All bias correction procedures in the manuscript were performed using a single iteration. But in terms of the flexibility of iROAR, we added an opportunity for the potential user to optimize the process of bias correction for their own multiplex system.

Line 87 "constant during a lifetime": this needs to be substantiated with either a reference or data. The authors show relevant data in Supplementary Figure 1, but there is no description of how the data shown in the figure were derived. For example: are all data points from a single study? do all data points used in a single regression (i.e., all those corresponding to the same gene segment) derive from the same patient? do all data in the figure derive from one single patient? what sequencing protocol was used? Has the proposed correction been applied? The strength of the claim should reflect the strength of the data. Note: I don't think the truth of this claim is a requirement for the proposed method to be valid. So if the data are such that the statement cannot be generalized to "all" repertoires, I don't see this as a problem.

We agreed with Reviewer 2, that this part is quite irrelevant to this study and has to be analyzed in more detail, so we deleted the statement "constant during a lifetime" along with Supplementary Figure 1.

Line 89 "reproducible": reproducible in what way? Over multiple aliquots of a sample? Over multiple samples for the same person? Over multiple people?

Reproducible in all these ways for each particular multiplex PCR primer system. The phrase “for the same multiplex PCR primer set” was added to the sentence.

Figure 1: Is the same stability observed for the sequencing protocols with even more amplification bias (VMPlex and VJMPlex)?

The point of the figure is that we show this stability on low-biased repertoires as a benchmark.

Lines 102 – 109: this paragraph is not clearly written, but it is the heart of the manuscript. The paragraph reads as if only the two terms with summations are based on out-of-frame rearrangements. I assume all four terms are based on out-of-frame rearrangements? Because otherwise the equation does not make sense …. Also, I recommend removing "a percentage of" in line 105, as this reads as if there is a percentage in the numerator of the numerator (i.e., the RC(Vi) term). Finally, in line 107, I assume you mean PCR amplification instead of clonal expansion?

Corrected

Figure 2: More information about the data is needed. Specific details are below.Lines 149 – 153 starting with "the average population frequencies": It isn't clear what is meant by "average population frequencies", and it isn't clear what the rest of the paragraph implies about calculations described in the paper and results displayed in the figures. It sounds as if different repertoires may have been subjected to different calculations.

“Average population frequencies” was replaced by “population frequencies”. The usage of population frequencies is optional. The users may need it when they face repertoires obtained using multiplex PCR with extremely broad primer efficacies. Analysis in the main text was performed without this option. Since *eLife* rules do not allow Supplementary notes, we decided to not discard this section from the paper to show to potential users the way of overcoming this possible pitfall.

Lines 157 – 158: The last sentence of this paragraph suggests that V- or J-genes with large or small relative frequencies may have been excluded. More details are needed. Were only large clones excluded? How were they identified for exclusion? By what method and what exclusion threshold was used? How many were excluded from each repertoire? If more than "a few", then before and after data need to be shown for some repertoires to show how this impacts the OAR distributions and averages.

In the current iROAR version user can define manually how many top non-functional clonotypes has to be excluded from OAR calculation. For most normal samples exclusion of one top clonotype for each V and each J-segment is sufficient to prevent inaccuracy in OAR calculation. To illustrate it we added suggested plots to Figure 4.

Lines 167 – 169: how is "adequate" defined? Within the 10% error range discussed in the context of Supplemental Figure 3?

Corrected:

“…adequate OAR values with the acceptable error rate of ~10%”.

Lines 257 – 262: These sentences suggest that the proposed approach is more reliable than biological spike-ins. This should be substantiated before being claimed. The authors state that this is because "the impact of CDR3 structure" is minimized. Why is this? Because the number of out-of-frame rearrangements is much much larger than the number of cell-line or synthetic spike-ins and so captures more CDR3 diversity? Or some other reason? More explanation of why this is being suggested should be given and the wording should make clear that this is an untested hypothesis.

We corrected the discussion sentence: “It makes this method potentially more reliable due to the ability to minimize the impact of CDR3 structure on PCR bias calculation since out-of-frame captures significantly higher CDR3 diversity”.